# A Methodological Approach to Assess the Effect of Organic, Biodynamic, and Conventional Production Processes on the Intrinsic and Perceived Quality of a Typical Wine: The Case Study of Chianti DOCG

**DOI:** 10.3390/foods10081894

**Published:** 2021-08-15

**Authors:** Francesco Maioli, Monica Picchi, Valentina Millarini, Paola Domizio, Gabriele Scozzafava, Bruno Zanoni, Valentina Canuti

**Affiliations:** Department of Agricultural, Food, Environmental, and Forestry Sciences and Technologies, University of Florence, Via Donizetti, 6-50144 Firenze, Italy; francesco.maioli@unifi.it (F.M.); monica.picchi@unifi.it (M.P.); valentina.millarini@unifi.it (V.M.); paola.domizio@unifi.it (P.D.); gabriele.scozzafava@unifi.it (G.S.); bruno.zanoni@unifi.it (B.Z.)

**Keywords:** Sangiovese, biodynamic wine, organic wine, wine quality, typicality, carbon footprint, polyphenols

## Abstract

The aim of this study was to propose a methodological approach to evaluate the impact of the organic, biodynamic, and conventional production processes on the intrinsic and perceived quality of a typical wine. For this purpose, fourteen commercial Chianti DOCG wines from the 2016 harvest were selected based on the type of production management. A survey was set up to get winemaking information from the estate’s producer of the wines to estimate the carbon dioxide production under the three types of management. The eligibility, identity, and style properties (the intrinsic quality) of the wines were defined. A group of 45 experts evaluated the differences between wines by the Napping test and rated their typicality (perceived quality). The organic and biodynamic management showed a lower level of estimated values of carbon dioxide production. The overall statistical elaboration of the chemical and sensory data highlighted that the registered differences of the intrinsic, perceived quality, and typicality level of the respective wines, did not depend on the type of management. The comparison of the three kinds of wine by SIMCA modeling, put in evidence that the conventional ones showed a greater homogeneity regarding chemical composition, sensory characteristics, and typicality.

## 1. Introduction

In recent years, in all agri-food sectors, consumers have shown they are increasingly sensitive to environmental issues, directing their portfolios accordingly [1,2,3]. Consequently, on the producer’s side, there has been a change of course from conventional practices of vineyard management and winemaking to more environmentally friendly techniques such as organic and biodynamic farming [4]. These two farming management types are closely connected; organic agriculture is regulated at the European level with a series of rules (EU Council Regulation EC No. 834/2007 and EC No. 203/2012); biodynamic, in contrast, considers organic management a prerequisite and applies voluntary regulation in keeping with the belief that a farm should be considered as a living organism. Demeter^®^ is the world-spread voluntary biodynamic certification and it provides the guidelines for the biodynamic winemaking process.

From surveys of the Research Institute of Organic Agriculture (www.fibl.org, accessed on 15 July 2021), which periodically investigates organic agriculture in the world, in 2018 it emerged that worldwide organic grape production increased by 422.300 hectares, a rise of 5% from the previous year.

In Italy, many vineries are converted, or are under conversion, to organic, and to a lesser extent, to biodynamic management, and, also for the realization of prestigious wines belonging to the Protected Designation of Origin (PDO) or the Italian DOC or DOCG.

The impact of organic and biodynamic vineyard practices on global wine quality has become a topic of great interest for many researchers. Many authors studied the effect of management systems on grapes and wine quality focusing on the chemical composition of wines, in particular, on the polyphenol content [5,6]. However, very few articles indicate how the different winemaking processes (conventional, organic, and biodynamic), could affect the quality of wine [7].

It has been reported that eco-certification is associated with the highest quality wines [8], even though this is not always perceived by consumers [9]. Compared to conventional wines, biodynamic ones have sometimes been reported as superior in quality [10,11] or similar [1,12].

No scientific paper regarding the impact of different production process approaches on the PDO wine typicality-expression is available.

In the PDO wines, more than others, the concept of quality is closely related to that of typicality. For the wine industry, regional typicality is an important concept as it delineates wines with recognizable sensory characteristics and chemical composition as well as the relative geographic areas [13,14,15]. In a PDO context, typicality is defined as the characteristics of a product from a *terroir*, meaning that the product is representative of its *terroir*. Thus, typicality can be defined as a set of properties of belonging and distinction [16] described by an intrinsic and perceived quality. Considering the absence of defects as a pre-requisite, some authors [17,18] proposed that the intrinsic quality is the result of three different profiles: (i) an eligibility profile, whose parameters are common to all wines (e.g., the sensory attributes and chemical compounds related to acidity, astringency, persistence, alcohol, viscosity, etc.); (ii) an identity profile, whose parameters are related to the grape variety and the *terroir* (aroma profile); (iii) a style profile related to the brand and expression of the winemaking. In keeping with the assumption that taste and tactile sensations do not allow for the recognition of a wine, the eligibility profile can change over time according to the market needs, without affecting the identity of the product [18]. On the contrary, the identity profile, given its connection with the recognizability elements (e.g., the typical volatile profile of the different grape varietals and their products), cannot change because it represents the distinct characteristics that define the identity and, consequently, the typicality of a wine [19]. Finally, the style profile can change over time as a function of the market and/or the winery brand needs, as long as it does not cover or alter the identity profile of the wine. The perceived quality is the result of the evaluation of the recognizability level and of its fitting with the identity model represented by a PDO. It is usually evaluated by experts since only subjects experienced with the product and its diversity (enologists, winemakers, sommeliers, etc.) can truly judge typicality to select the most representative one [19]. In fact, the typicality concept is supported by the existence of a common memorized prototype that represents the image of all the previous experiences of wines of that type [14,20,21].

Starting from the notion that biodynamic provides a “soft” approach in which human operations are reduced to a minimum, several biodynamic producers consider their wines as more respectful of the characteristics of the raw material and, given the correlation of typicality to those characteristics and to the *terroir*, they consider their products more typical than the conventional and the organic ones [21,22].

The aim of this study was to propose a methodological approach to evaluate the impact of different production processes (conventional, organic, and biodynamic) on the intrinsic and perceived quality of a PDO wine. For this purpose, a case study of Chianti DOCG was set up selecting commercial wines from conventional, organic, and biodynamic wineries. Wines from the 2016 harvest were chemically and sensorially analyzed in order to put in evidence correlations between the different winery management types and the quality of the relative wines.

## 2. Materials and Methods

### 2.1. Wine Samples

Fourteen commercial Chianti DOCG, Sangiovese-based wine, of three different estate management types (conventional, organic, and biodynamic), from the 2016 harvest were selected among wineries located in a limited Chianti area (Table 1) to collect wines produced in similar environmental conditions. The prerequisite for the present study was that all wines were approved as Chianti DOCG. Hence, it was selected a set of commercial wines that had passed a chemical and sensory examination by a specific commission for the achievement of Chianti DOCG eligibility. A minimum of 12 bottles was received for each wine sample and they were stored in the university underground cellar at a temperature of −18 °C until the analysis.

### 2.2. Vineyard and Winemaking Procedures Survey

The winegrowers were asked to participate in a survey to achieve the information related to the production process (vineyard and winemaking management and practices). Given the oenological aim of the present study, data relative to vineyard procedures (e.g., vine yield, pest control, fertilization, and harvest method) were collected only for an informative scope in order to complete the overview of the different production processes. Concerning the cellar procedures, the relative value of the carbon footprint, expressed in Kg CO_2_ eq/ton, was associated with the category and quantity of products involved in the winemaking process. The estimation of the CO_2_ emission values was obtained by previous research in this field [23,24] and related to the production of 1 hL of wine. In particular, the following values were associated with the different oenological practices (expressed in Kg CO_2_ eq/ton of product): tartaric acid 3300, potassium disulfite 1470, tannins 220, ammonium sulfate and enzymes 733, liquid SO_2_ 440, yeast and cells extract 2200, and Arabic gum 400.

### 2.3. Intrinsic Quality: Chemical Characteristics for Measuring Eligibility, Identity, and Style Wine Properties

The eligibility chemical characteristics were represented by standard parameters, polyphenol profiles, and color indices; the identity chemical characteristics were represented by the volatile fractions of the wines [17,18]; the style requirement was represented by the chemical variables related to wine aging (e.g., vanillin and γ-lactones). The standard parameters (pH, titratable acidity, volatile acidity, alcohol content, residual sugars, SO_2_ content) were measured according to the official EU methods (Official Methods of Wine Analysis, Reg. 440/2003). Regarding the phenolic profile of wines, monomers, and polymerized anthocyanins (polymeric pigments) (both expressed as mg/L of malvidin-3-*O*-glucoside), and tannins (expressed as mg/L (+)-catechin) were measured by HPLC [25,26] and was carried out on a Perkin Elmer Series 200 LC equipped with an autosampler and a diode-array detector (Perkin Elmer, Shelton, CT, USA). Chromatograms were acquired at 280 nm and 520 nm, recorded and processed using the Total Chrome Navigator software (Perkin Elmer). Color intensity (CI) and hue (Hue) values were measured according to the method of Glories [27], the total phenols index (TPI) was measured as described by Ribereau-Gayon [28], and the gelatin index (GI) according to Mirabel [29]. The ultraviolet–visible (UV–Vis) absorbance of the samples was measured on a Perkin Elmer Lambda-35 UV–Vis Spectrophotometer (Perkin Elmer). Milli-Q water was used as a reference (>18 MΩ·cm, Milli-Q Element system, Millipore, Bedford, MA, USA). CIE (Commission Internationale de l’Eclairage) L*, a*, and b* color coordinates were also measured. Visible spectra were recorded at 400–700 nm reflectance using the same spectrophotometer equipped with the RSA-PE-20 Integrating Sphere accessory assembly (Labsphere, North Sutton, NH, USA). UV WinLab Software was used to record the spectra (version 2.85.04, Perkin Elmer) and CIE L*a*b* color coordinates were calculated using Color software (version 3.00, 2001, Perkin Elmer). All the wine samples were analyzed in triplicate.

Regarding the volatile compounds, higher alcohols and ethyl aldehyde were determined using an AutoSystem XL gas chromatograph equipped with FID (flame ionization detector) (Perkin Elmer), according to the method developed by Bertuccioli et al. [30]. The volatile compounds were expressed as mg/L using a calibration curve obtained by the injection of the reference compounds (≥99% purity) purchased by Sigma (Sigma-Aldrich, Saint Louise, MO, USA). The free volatile profile was determined by the HS-SPME/GC–MS method developed by Canuti et al. [31]. The analytical system for the determination of the volatile compounds comprised an AutoSystem XL gas chromatograph (Perkin Elmer) paired with a Turbomass Gold mass selective detector (Perkin Elmer). The software used was TurboMass v.5.1.0. An Innowax column (30 m × 0.25 mm o.d., 0.25 μm film thickness, Agilent Technology, Folsom, CA, USA) was used. Volatile compounds were identified and quantitated by using the reference compounds (≥99% purity) purchased by Sigma-Aldrich. All the compounds for which the reference standards were not available were quantitated based on the relative response to the octan-2-ol internal standard. All the wine samples were analyzed in triplicate.

### 2.4. Intrinsic Quality: Sensory Attributes Measuring Eligibility, Identity, and Style Wine Properties

A panel of trained judges carried out the sensory evaluation according to the Quantitative Descriptive Analysis (QDA) method. Seventeen trained judges (eleven males and six females), recruited from students, staff, and friends of the DAGRI Department in Florence formed the panel. The panel was already trained for the evaluation of the red wine and was submitted to further training for the set of Sangiovese during six sessions. In the first session the judges tasted and described the taste and tactile descriptors of the wines, while in the three subsequent sessions, they described and discussed the volatile profile. In every session, the panel was provided with a set of reference standards, prepared as illustrated in Table 2. At the end of the training sequence, two sessions of trial evaluation were performed.

All of the sensory evaluations were performed in isolated and ventilated sensory booths under red lights to eliminate bias attributed to color differences. The presentation was monadic with a balanced presentation order for carry-over effect, according to an uncompleted block design, with seven wines per session evaluated in three replications, for a total of six sessions. The wine samples (30 mL) were poured at room temperature (approximately 19 °C) and presented in standard tasting glasses (ISO-3591, 1977) covered with plastic lids and identified by random three-digit codes. The sensory data from the six sessions’ descriptive analyses were combined using the shared or synonymous attributes and standardized to mean zero for each sensory attribute within each descriptive analysis. In each step, the samples were evaluated globally (i.e., ortho-nasal aroma after swirling, plus retro-nasal aroma, taste, and mouthfeel after sipping). After every sample, the judges had to wait for 60 s, during which time they were asked to rinse the mouth with water. All of the samples were expectorated. Every evaluation session lasted approximately 15 min. The panelists answered on a 10-point category scale (one scale per sample), anchored with 1 (absent) on the left end and 10 on the right end (very strong). The reference standards submitted to the judges corresponded to 6 on the intensity scale (medium intensity).

According to Bertuccioli et al. [17] and Canuti et al. [18], the eligible sensory profile was described by the following attributes: acidity, sweetness, bitterness, and astringency. The identity profile was defined by the following aromatic attributes: blackberry, prune, cherry, floral, vegetal, vanilla, and spicy odor, fruity, floral, spicy, and vegetal in-mouth flavor. One attribute described the style requirement, that is wood odor and in-mouth flavor.

All of the sensory data were collected using FIZZ software (Version 2.00L, Biosystemes, Couternon, France). The sensory evaluation took place in the wine sensory laboratory at the DAGRI, the University of Florence in Florence (Italy).

### 2.5. Perceived Quality: Napping^®^ Test and Wine Rating of Typicality

Forty-five wine experts (producers, oenologists, winemakers, and sommeliers) were selected based on their extensive experience with Chianti DOCG wine, performed the Napping test and the typicality evaluation. Napping [32] is a specific variant of Projective Mapping, a method originally proposed for applied sensory studies by Risvik et al. [33] to describe overall differences among samples. All of the sensory evaluations were performed as previously described for the evaluation of the intrinsic quality. For both tests (Napping and typicality), a complete randomized and balanced experimental design was followed for the presentation order.

In the Napping session, the 14 samples were simultaneously presented to the judges, who were required to place them on a two-dimensional space (blank piece of white paper with dimensions 60 cm × 90 cm), in a way that reflected their perceived sample differences, i.e., by placing samples perceived as similar close to each other, and samples perceived to be different further apart. The data from the Napping test were digitized by writing in a table, for each product, its X-coordinate and Y-coordinate on the sheet. The origin was placed on the left bottom corner of the sheet.

After finishing the Napping test, a new set of the same samples, with a different presentation order, was presented to the panelists who were instructed as follows [34]:
*“Imagine that you want to explain to someone what a Chianti DOCG wine is. To explain, you can suggest to this person to taste a wine. For each wine presented, you must answer the following question: Do you think that this wine is a good example or a bad example of what a Chianti DOCG wine is?”*

The score of every sample was assigned on a categorical scale, from 1 to 7, anchored at left to “*very bad example*” and on the right to “*excellent example*”. In each step, the samples were evaluated globally (i.e., orthonasal aroma after swirling, plus retronasal aroma, taste, and mouthfeel after sipping). Water was provided as a palate cleanser. The whole session lasted approximately 20 min.

### 2.6. Statistical Analyses

The data sets of the wines were analyzed by analysis of variance (ANOVA), considering “estate management” and “replicates” for the chemical data, and “estate management”, “replicates”, and “judges” for the sensory data as factors, using Statgraphics Centurion (Ver.XV, StatPoint Technologies, Warrenton, VA, USA). Based on the previous ANOVA, a Fisher’s LSD post hoc test was used to determine the significant differences between group means (*p*-value = 0.05).

Principal Component Analysis (PCA) and Multiple Factor Analysis (MFA) (Escoffier and Pagès 1994, Pagès and Husson 2001) in which each subject of the Napping panel (experts) constitutes a group of two un-standardized variables was performed using XLSTAT v. 2020.5.1 (Addinsoft, Paris, France). The typicality scores of the second table (45 columns) were considered as a set of 45 supplementary variables: they did not intervene in the axes construction, but their correlation coefficients with the factors of MFA were calculated and represented as in a usual PCA.

For the evaluation of the differences between wines, three different global models were created (conventional, organic, and biodynamic) using a Soft Modelling of Class Analogy (SIMCA) performed using Unscrambler (V9.1, CAMO Process AS, Oslo, Norway). Within each model, the variables involved have been separated on the basis of the chemical and sensory eligibility and identity profile. The construction of the models was carried out as follows:

*Phase 1.* Three models were built to describe each type of management. In detail, for each type of management (organic, biodynamic, and conventional) the models were built for chemical and sensory eligibility and identity variables;

*Phase 2.* The fitting (classification) of organic, biodynamic, and conventional wines into the three management wine models previously created.

## 3. Results and Discussion

### 3.1. Vineyard and Winemaking Procedures and Carbon Emission Estimation

The data collected by the survey submitted to the winegrowers were resumed in Table 3 to highlight the main differences between the three different approaches. The organic and biodynamic vineyards were treated by applying only sulfur-copper-based compounds for pest control, while the conventional ones were allowed to use synthetic phytosanitary products as well as sulfur-copper-based compounds. There were also differences in soil management such as fertilization: the most common practices in organic and biodynamic management were green manure and compost application, while in the conventional one, both compost and mineral NPK fertilizers were applied.

All the wines, except for E_OR, F_OR and G_OR, were Sangiovese-based plus a percentage of a blend of different red grape varieties (for the Chianti DOCG, Sangiovese grape should be from 70% to 100% and from 30% to 0% for other red grape varieties admitted by the production requirements).

During the winemaking process, several coadjuvants, additives, and techniques can be applied in order to perform the must fermentation, wine clarification, stabilization, till bottling. Usually, given their official set of rules (EU Council Regulation EC No. 834/2007, EC Reg No. 203/2012 and Demeter protocol) biodynamic and organic winemaking processes are characterized by less addiction of sulfites (less or equal to 70 mg/L for biodynamic, 100 mg/L for organic, and 150 mg/L for conventional) and other additives, with the biodynamic more restrictive than the organic. This was confirmed in this case study, as evidenced in Table 3 and Figure 1: biodynamic and organic farmers, except for one organic producer (F_OR), did not inoculate selected yeasts to conduct alcoholic fermentation; biodynamic wineries (C_BD and D_BD) did not apply temperature control, some biodynamic and organic wineries (D_BD and E_OR) did not perform product addiction and sterile filtration. Instead, the conventional wineries involved in the present study adopted most of the allowed oenological practices and products.

The data obtained from the survey related to the winemaking practices were converted into carbon footprint values to compare the three estate management types based on their carbon emission production. Figure 1 resumes the impact in terms of kg CO_2_ eq per hL of wine related to winemaking practices. In detail, organic wineries ranged from 0.00588 to 0.30768 kg CO_2_ eq/hL, biodynamic from 0.00882 to 0.04296, while conventional had values ranging from 0.144 to 0.699979 kg CO_2_ eq/hL. In the present case study, the higher values of the conventional management can be due to the general trend of using a higher amount of admitted products together with different oenological practices (temperature control, selected yeast, enzymes, tannins, nutrient salts, yeast extract, SO_2_, tartaric acid, and sterile filtration).

### 3.2. Evaluation of the Chemical and Sensory Intrinsic Quality of 2016 Chianti Wines

#### 3.2.1. Chemical Analysis

Two-way ANOVA was carried out on wine chemical parameters using estate management and analysis replicates as factors. Replicates showed no significance. The wine G_OR was identified by the Q test as an outlier, so this wine was excluded by all the statistical elaboration carried out.

Table 4 reported mean values and significance between different estate management types for eligibility and identity variables, respectively. Table 4 showed that organic wines had a lower pH, higher values of color indices (TPI, CI), gallic acid, quercetin-3-*O*-glucoside, and together with the biodynamic wines, the higher values of gelatin index, the content of tannins, alcohol, and polymeric pigments. Conventional wines, instead, had a larger amount of total sulfur dioxide, higher values of L*, hue, and pH, and lower content of tannins, polymeric pigments, and alcohol. According to these results, some differences emerged in relation to the estate management and, in particular, among wineries due to different oenological practices that can be adopted during the winemaking process. In more detail, organic wines showed higher phenolic compound content, and consequently, higher color intensity. In the present study, biodynamic wines were usually less rich in total anthocyanins and quercetin. This was in agreement with other authors that found that biodynamic wines were usually less rich in total anthocyanins and quercetin [12,35]). In this case, biodynamic wines showed lower amounts of monomer anthocyanins such as petunidin-3-*O*-glucoside, and a higher content of polymeric pigments, tannins, and *trans*-caftaric acid. Consequently, organic and biodynamic Chianti wines showed, as evidenced by other authors [7], a more evolved and stable color due to the higher content of polymeric pigments than the conventional ones, which instead showed higher content of monomer anthocyanins. The chemical identity profile of the Sangiovese wines was represented by the volatile compounds originating starting from the grape and by the alcoholic and malolactic fermentation (i.e., terpenes, norisoprenoids, acetates, esters, acids, alcohols). Results reported in Table 4 highlight conventional wines as having a higher quantity of acetaldehyde and lower content of ethyl acetate than organic and biodynamic wines. Similar results were obtained by Picchi et al. [7], according to which, the higher content of acetaldehyde and monomer anthocyanins and the lower amounts of polymeric pigments in conventional wines were symptomatic of less reactivity of these wines. Organic and biodynamic wines had the highest content of ethyl acetate, diethyl succinate, and isoamyl acetate. Conventional wines had a larger amount of 2-octanone together with the organic wines and octanoic acid like the biodynamic wines. Other authors [12] reported the composition of Sangiovese wines, obtained with different estate management, as having a tendency (but not significant) for biodynamic wines to contain more diethyl succinate, a trend that in the present study on Chianti DOCG 2016 turned out to be significant (F-value 4.03*).

A PCA was run to obtain a spatial distribution of wines for significant chemical parameters (eligibility and identity variables) (Figure 2a,b). The data obtained from the surveys related to the winemaking process, converted into carbon footprint values to compare the three estate management types based on their carbon emission production (Figure 1), were then plotted on the chemical variables as supplementary data [32] (Figure 2b). Wines were separated in a two dimensions map, according to the type of estate management (46.97% of total explained variance) (Figure 2a,b) as follows: conventional wines were grouped on the left side of the graph and organic and biodynamic on the right side. In particular, the wines on the left side, most of the conventional ones and the organic I_OR and F_OR, were correlated with all the winemaking supplementary variables considered, such as the use of selected yeast, bottling treatments (sulfites, Arabic gum, tartaric acid, filtration), and addition of coadjuvants (e.g., tannins, yeast extract) and additives (e.g., NH_4_ salts, sulfites) indicating an application of admitted enological practices with a larger impact on the CO_2_ emission. Moreover, it is possible to evidence further subgroups such as the wines B_BD, C_BD, and D_BD that were characterized by caffeic acid, isoamyl acetate, total tannins, and polymeric pigments. A second group was represented by the wines A_BD, E_OR, and H_OR characterized by CI, TPI, alcohol, total acidity, quercetin-3-*O*-glucoside, and gelatin index. Another subgroup could be evidenced on the left side of the graph constituted by the wines M_CV and N_CV related to the variables pH, octanoic acid, and L*. O_CV wine was defined by the variables petunidin- and peonidin-3-*O*-glucoside, acetaldehyde, procyanidin B1, and total SO_2_. The wines F_OR, I_OR, L_CV, and P_CV results were the least explained by the significant variables.

#### 3.2.2. Sensory Analysis

The biplot in Figure 3 shows the relationship between the wine samples and the significant descriptive attributes (79.45% of total explained variance), with estate management, judges, and replicates as factors. It is possible to see that the samples were separated into two main groups along the first dimension: N_CV, P_CV, C_BD, M_CV, E_OR, H_OR, and, even if to a lesser extent, D_BD, on the left side and I_OR, L_CV, B_BD, O_CV, F_OR, and A_BD on the right side. The two groups were related to the attributes Vanilla Flavor, Wood Odor, Astringency, and Bitterness on the left side, and Vegetal Odor and Flavor on the right side. The groups were further divided along the second dimension, with the samples N_CV and P_CV related to the attribute Vanilla Flavor and E_OR, H_OR, and D_BD to the attributes Wood Odor, Astringency, and Bitterness. Samples C_BD and M_CV were equally characterized by all the aforementioned attributes. When data were analyzed for the factors sample instead of estate management, the wines C_BD and M_CV were also characterized by fruity attributes (Fruity Flavor, Prune, and Blackberry Odor) (data not shown). On the right side, the samples I_OR, L_CV, B_BD, and even if to a lesser extent, A_BD and F_OR, were related to the attributes Vegetal Odor and Flavor. The sample O_CV seemed to not be directly related to any of these two significant attributes, but when data were analyzed by the factor sample instead of estate management, this wine was related to the attribute Floral Flavor and Acidity (data not shown).

The descriptors Astringency and Bitterness seemed to be related to the eligibility chemical parameters that described the same wines (E_OR, H_OR, and D_BD) since they were richer in polyphenols.

#### 3.2.3. Chianti DOCG Wines Classification according to Estate Management Models

To better explain the similarities and differences that emerged between the wines by the PCA, eligibility and identity chemical and sensory wine profile models were built for every management type (conventional, organic, and biodynamic) using the SIMCA. The results reported in Table 5 allowed the classification of the wines as a function of their fitting in a different kind of model previously built with SIMCA. It is possible to evidence that the conventional model was the most restrictive since several organic and biodynamic wines did not fit with it, in particular, for the eligibility and identity sensory variables. In contrast, the organic model allowed for the fitting of almost all the wines with two exceptions (C_BD, N_CV). The biodynamic model allowed for the fitting of all the conventional and organic wines resulting in it being the least restrictive. The level of restriction of the SIMCA models could be translated in this case study in terms of the quality control level of the production chain. In fact, the looser the model, the greater the variability of the samples were, which in turn, was related to the quality characteristics of a certain type of wine. This depended on the analyzed biodynamic wines that, more than the organic and conventional, included both the highest and the lowest quality level wines, and as consequence, very different levels of typicality. Moreover, the conventional wines were more homogeneous for the quality level with more similar typicality scores.

### 3.3. Evaluation of the Sensory Perceived Quality of 2016 Chianti Wines: Napping and Typicality

The graphic in Figure 4a,b showed the results of the Napping test and the typicality evaluation (35.39% of total explained variance). It is possible to see that samples were separated along the first dimension in three main groups (Figure 4a): the samples F_OR, B_BD, A_BD, and D_BD on the left side; the samples I_OR, L_CV, H_OR, E_OR M_CV, and O_CV in the middle area; the samples C_BD, N_CV, and P_CV on the right side of the plot. The samples in the middle area are less defined due to higher variability of their position in the single maps. The conventional samples are grouped on the right side, with the exception of the sample L_CV that, given its position in the biplot, is also less defined. These wines are more distributed along the second dimension, which accounts for less explained variance (14.95%). The position of organic and biodynamic samples, given their deployment along the first dimension, explains most of the variance and shows more variability. The results of the Napping test can be translated in terms of the level of quality as it was quite coherent with the disposition of the samples obtained by the QDA analysis: the N_CV, P_CV, and C_BD were grouped and related to the descriptor Vanilla, in opposition to several of the organic and biodynamic samples such as I_OR and B_BD which were related to the descriptor Vegetal. The consensus map of the Napping was confirmed by the evidence resulting from the SIMCA analysis, which could be considered the expression of the intrinsic quality of the different kinds of wines. In fact, both the results highlighted that the biodynamic wines were the most spread in terms of their sensory characteristics compared to the conventional and organic wines. Figure 4b shows the distribution of the typicality expert scores of the wines. The distribution of the scores evidenced a clear trend to reward the samples on the right side of the Napping map (C_BD, N_CV, P_CV, M_CV), with a few of the experts rewarding the samples of the left side (H_OR, F_OR, D_BD, A_BD).

Table 6 shows the mean values of typicality scores assigned to the wines by the experts. Statistical differences were highlighted by the analysis of variance between the three different estate management types for the typicality scores: the biodynamic wines obtained the lower mean score, even if the most rewarded was a biodynamic one (C_BD), while the conventional obtained the highest. The analysis of the results showed that for every estate management type there was at least a sample that obtained a high typicality score (C_BD, P_CV, and E_OR) confirming that, as in previous studies [12,35,36], the type of estate management could not be considered, in itself, an element of quality.

This evidence is better explained in Figure 5, which reports the boxplot of the typicality scores, grouped by estate management. The biodynamic wines (BD) showed wider extremes compared to organic and conventional wines. Specifically, biodynamic wines obtained a minimum of 3.2 and a maximum of 5.73, with higher variability (SD = 1.1). Organic (BD) and conventional (CV) wines seemed to have more standardized sensory characteristics, affected by less variability: they obtained, respectively, a minimum of 4.16 (OR) and 4.64 (CV), and a maximum of 5.29 and 5.58, with a lower standard deviation of 0.49 and 0.39, respectively.

## 4. Conclusions

This study aimed to propose a methodological approach to evaluate the impact of different types of vineyard management and oenological processes on the intrinsic and perceived quality of a typical wine in terms of eligibility and identity profile. This was achieved by comparing a homogeneous group of 2016 Chianti DOCG commercial wines obtained according to conventional, organic, and biodynamic production processes.

The results showed that the wines did not present systematic differences in the eligibility and identity profiles according to the type of production process. However, some significant differences were found between the values relating to the phenolic composition and the volatiles. In particular, organic and biodynamic wines showed higher values of color intensity, total polyphenol index, and polymeric pigments at the expense of monomer anthocyanins. However, a higher content of diethyl succinate and isoamyl acetate was found in the biodynamic wines and of acetaldehyde in the conventional ones.

From the sensory point of view, the set of analyzed samples did not present systematic differences related to the type of estate management, even if significant differences were detected for several attributes. The eligibility profile attributes of Astringency and Bitterness were confirmed as being related to the chemical parameter’s total phenols index and gelatin index. In general, it was possible to confirm that organic and biodynamic wines resulted in being more evolved in terms of color stability compared to the conventional ones.

The SIMCA model built on the chemical and sensory profiles highlighted that the conventional wine model presented less variability, as opposed to the biodynamic model with results that were more variable in terms of intrinsic and perceived quality.

The expert scores highlighted that the estate management process could have a different effect on typicality, but also that every type of management could achieve a wine in keeping with the reference frame of typicality.

The comparison between the organic, conventional, and biodynamic production processes of typical wines, therefore, offers interesting ideas for further future investigations. Given today’s knowledge about the intrinsic quality of wine and based on the survey conducted, for the first time in this study about perceived quality, it is possible to state that less impact in terms of carbon dioxide emission during the winemaking process, does not represent an obstacle, in itself, to the production of a typical wine. This evidence supports, however, that in order to produce a typical wine, controlling the process represents the critical point for any type of management.

## Figures and Tables

**Figure 1 foods-10-01894-f001:**
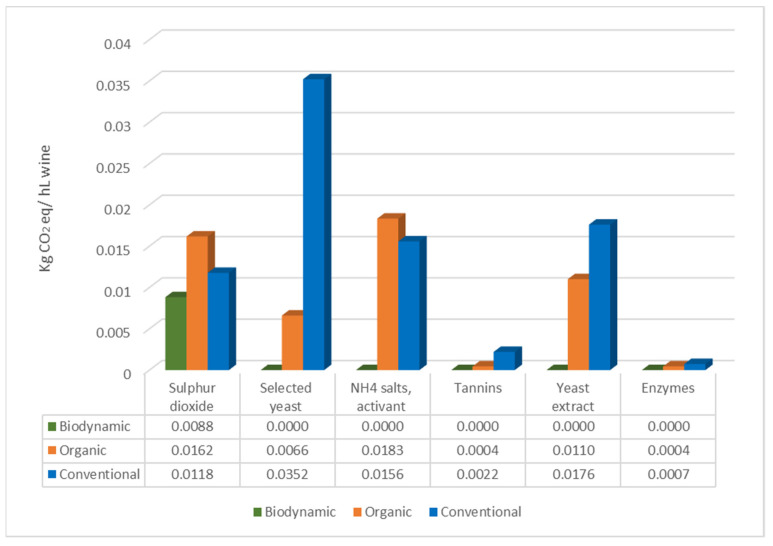
Estimated carbon dioxide emission expressed as kg of CO_2_ eq/hL of wine for biodynamic, organic, and conventional wineries based on the survey data (average values).

**Figure 2 foods-10-01894-f002:**
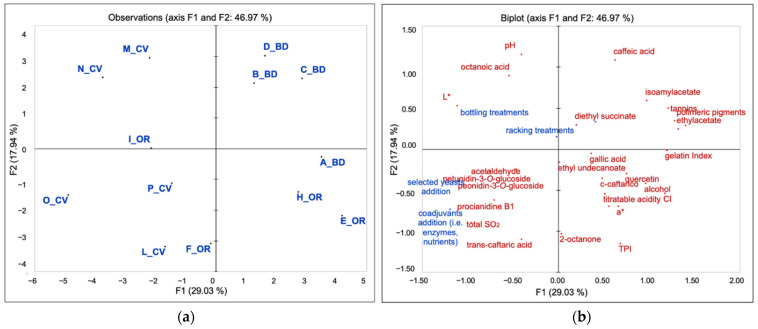
Principal Component Analysis (PCA) of 2016 Chianti wines (**a**) with the significant chemical variables and the survey data (related to the winemaking process) elaborated as supplementary (**b**).

**Figure 3 foods-10-01894-f003:**
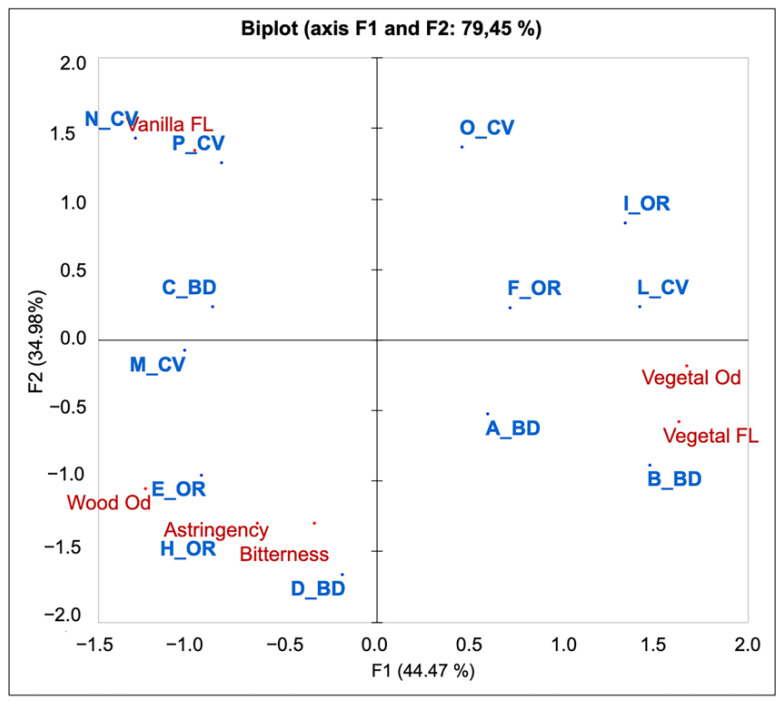
Principal Component Analysis (PCA) of Chianti 2016 wines with the significant sensory variables for the factor “estate management”.

**Figure 4 foods-10-01894-f004:**
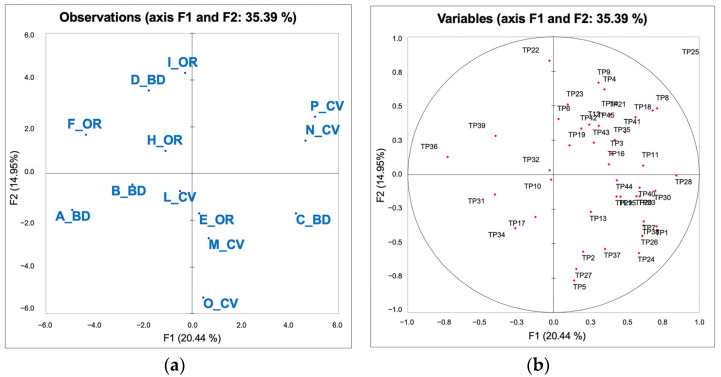
Representation of the Chianti DOCG wines by multiple factor analysis according to the Napping X- and Y-coordinates and typicality scores provided by the panel of experts. (**a**) Wine distribution; (**b**) distribution of the typicality scores (elaborated as supplementary data).

**Figure 5 foods-10-01894-f005:**
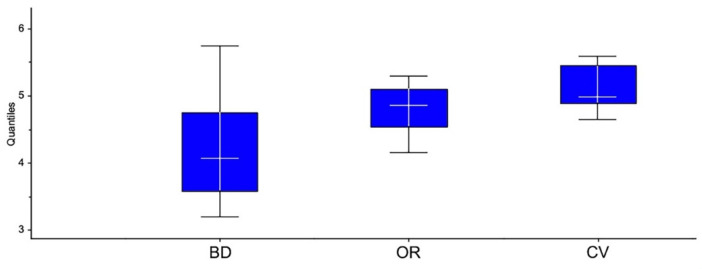
Typicality scores of Chianti DOCG wines from different estate management types (BD biodynamic, OR organic, CV conventional): minimum and maximum, 25% and 75% percentiles, and median. Mean values and standard deviation (with significance *p*-value < 0.001: BD, 4.27a ± 1.10; OR, 4.79b ± 0.49; CV, 5.11b ± 0.39).

**Table 1 foods-10-01894-t001:** Winery code, vineyard, and winery management, and geographical area coordinates.

Winery Code	Management of Vineyard and Winery	Geographical Coordinates
A_BD	biodynamic	43°24′33.6″ N 11°08′41.9″ E
B_BD	biodynamic	43°46′30.0″ N 10°53′25.4″ E
C_BD	biodynamic	43°40′56.3″ N 10°53′23.2″ E
D_BD	biodynamic	43°36′29.1″ N 11°09′57.4″ E
E_OR	organic	43°43′26.9″ N 11°03′09.6″ E
F_OR	organic	43°38′41.4″ N 11°02′34.7″ E
G_OR	organic	43°42′26.8″ N 11°10′57.9″ E
H_OR	organic	43°32′24.9″ N 11°08′58.9″ E
I_OR	organic	43°45′16.6″ N 11°02′37.9″ E
L_CV	conventional	43°34′12.2″ N 11°10′53.4″ E
M_CV	conventional	43°31′24.0″ N 11°08′47.5″ E
N_CV	conventional	43°39′43.2″ N 11°03′26.9″ E
O_CV	conventional	43°37′54.1″ N 11°07′15.4″ E
P_CV	conventional	43°38′33.0″ N 11°02′13.0″ E

**Table 2 foods-10-01894-t002:** Attributes and Reference Standard for the panel training.

Category	Attribute	Reference Standard
FRUITY Smell and flavor in mouth	Blackberry Jam	1 g blackberry jam in 1 mL of base red wine
	Prune	1 mL prune syrup juice in 1 mL of base red wine
	Cherry	0.5 mL prune cherry juice in 1 mL of base red wine
FLORAL Smell and flavor in mouth	Floral	1 mL stock solution 1 + 1 mL stock solution 2 * in 2 mL of base red wine
VEGETAL Smell and flavor in mouth	Cooked Vegetal	0.5 mL canned asparagus juice in 1 mL of base red wine
WOOD Smell and flavor in mouth	Wood	1 mL stock solution 4 ** in 1 mL of base red wine
	Spicy	1 mL stock solution 5, 6, and 7 *** in, respectively, 1 mL of base red wine
TASTE	Acid	0.04 g citric acid in 100 mL of base red wine = intensity 9
	Sweet	0.8 g of glucose in 100 mL of base red wine = intensity 9
TACTILE SENSATIONS	Astringency	0.3 g potassium alum in 100 mL of base red wine = intensity 9

* Wild iris petals (stock solution 1), rose petals (stock solution 2) in 100 mL of base red wine for one hour; ** 10 g of chips in 100 mL of base red wine for 2 h; *** 1 g of black pepper crushed grains (stock solution 3), 1 g crushed cloves (stock solution 4), 1 g pieces of cinnamon (stock solution 5) in, respectively, 100 mL of base red wine for 30 min; reference standard were daily prepared. Stock solutions were prepared every 2 weeks and stored at 4 °C.

**Table 3 foods-10-01894-t003:** Survey overview: vineyard and winemaking procedures and products used in the different estates (A-P: estate; BD, biodynamic, OR, organic, and CV, conventional management).

Estate Code	A_BD	B_BD	C_BD	D_BD	E_OR	F_OR	G_OR	H_OR	I_OR	L_CV	M_CV	N_CV	O_CV	P_CV
**Vineyard**														
Kg/Plant	n.a. ^5^	1.70	1.4	1.2	2.57	2.18	1.57	1.2	1.2	1.57	2	1.83	n.a. ^5^	1.6
Harvest	n.a. ^5^	manual	manual	manual	manual	manual	manual	manual	mechanic	manual	manual	mechanic	n.a. ^5^	mechanic
Fertilization compounds	n.a. ^5^	green manure	green manure	green manure	n.a. ^5^	green manure, organic	green manure, mineral NPK	green manure	n.a. ^5^	organic, mineral NPK	mineral NPK	organic, mineral NPK	n.a. ^5^	mineral NPK
Pest control compounds	n.a. ^5^	Cu ^6^ S ^7^ Bt ^1^	Cu ^6^ S ^7^ Bt ^1^	Cu ^6^ S ^7^ Bt ^1^	Cu ^6^ S ^7^	Cu ^6^ S ^7^ Sp ^2^	Cu ^6^ S ^7^	Cu ^6^ S ^7^	n.a. ^5^	n.a. ^5^	Cu ^6^ S ^7^ Sint ^8^	Cu ^6^ S ^7^ Sint ^8^	n.a. ^5^	Cu ^6^ S ^7^ Sint ^8^
**Cellar**														
Tank material	n.a. ^5^	SS ^4^	CR ^3^	SS ^4^	CR ^3^	SS ^4^	SS ^4^	SS ^4^	SS ^4^ CR ^3^	CR ^3^	SS ^4^	CR ^3^	SS ^4^	SS ^4^
Tank volume (hL)	n.a.^5^	65	120	80	30	150	100	50	100	150	100	120	1500	150
Temperature Control	n.a. ^5^	n.a. ^5^	-	-	n.a. ^5^	n.a. ^5^	√	n.a. ^5^	n.a. ^5^	-	n.a. ^5^	√	√	n.a. ^5^
SO_2_ at pressing ^9^	n.a. ^5^	4	-	4	4	5	√	6	√	3	n.a.^5^	5	8	10
Selected Yeast ^9^	n.a. ^5^	-	-	-	-	15	-	-	-	15	15	15	15	20
Blend	n.a. ^5^	√	√	√	-	-	-	√	√	√	√	√	√	√
Tannins ^9^	n.a. ^5^	-	-	-	-	10	-	-	-	10	√	5	15	10
NH_4_ salts ^9^	n.a. ^5^	-	-	-	-	100	-	-	√	20	√	30	10	25
Yeast extract ^9^	n.a. ^5^	-	-	-	-	20	-	-	√	-	-	30	10	-
Albumin ^9^	n.a. ^5^	-	-	-	-	-	-	√	-	-	-	-	-	-
Enzimes ^9^	n.a. ^5^	-	-	-	-	3	-	-	-	3	-	-	-	2
SO_2_ ^9^	n.a. ^5^	4	-	-	-	5	-	-	-	-	-	-	-	√
Filtration (µm)	n.a. ^5^	-	5	-	-	0.65	√	5	-	1	√	0.45, 1	√	1
SO_2_ at bottling ^9^	n.a. ^5^	√	-	2	-	5	-	50	-	-	√	1.5	√	√
Arabic gum ^9^	n.a. ^5^	-	-	-	-	-	-	√	-	-	√	-	√	√
Tartaric acid ^9^	n.a. ^5^	√	-	-	-	√	-	-	-	-	√	-	√	√

^1^*Bacillus thuringensis*; ^2^ Spinosad; ^3^ Concrete; ^4^ Stainless Steel; ^5^ Not available;^6^ Copper-based compounds; ^7^ Sulfur; ^8^ Synthetic pesticides; ^9^ g/hL.

**Table 4 foods-10-01894-t004:** ANOVA table of the significant eligibility and identity profile variables of Chianti 2016. Data are expressed as mean of three determinations; in each row, different letters (a–c) indicate statistically significant differences. F-values and significance, and standard deviation are also reported.

Eligibility Profile Variables	Biodynamic	Conventional	Organic	F-Value
Alcohol (% *v*/*v*)	13.93 ± 0.01 b	13.52± 0.01 a	14.05± 0.01 b	9.27 ***
pH	3.56 ± 0.00 b	3.56 ± 0.01 b	3.31 ± 0.01 a	8.12 **
Total SO_2_ ^3^	23.58 ± 0.58 a	56.07 ± 0.58 c	37.42 ± 0.46 b	20.59 ***
Total Phenol Index	50.66 ± 0.27 a	50.11 ± 0.12 a	59.52 ± 0.14 b	10.22 ***
Color Intensity	7.83 ± 0.01 a	6.62 ± 0.01 a	8.00 ± 0.01 b	13.85 ***
L*	78.22 ± 0.06 b	81.46 ± 0.03 b	77.96 ± 0.07 a	13.73 ***
a*	21.19 ± 0.07 a	18.25 ± 0.03 a	21.57 ± 0.03 b	7.01 **
Gelatin Index	49.57 ± 0.93 b	45.23 ± 0.37 a	48.94 ± 0.58 b	3.91 *
Gallic acid ^3^	53.18 ± 1.22 a	63.83 ± 4.68 ab	71.59 ± 5.41 b	3.56 *
Procyanidin B1 ^3^	35.02 ± 6.62 a	40.77 ± 4.68 a	52.28 ± 5.41 b	8.92 ***
Quercetin-3-*O*-glucoside ^3^	57.27 ± 9.42 a	59.54 ± 13.09 a	86.64 ± 10.64 b	4.29 *
*Cis*-caftaric acid ^3^	4.00 ± 0.29 ab	3.20 ± 0.11 a	4.42 ± 0.11 b	3.87 *
*Trans*-caftaric acid ^3^	14.12 ± 0.59 a	34.29 ± 1.49 b	40.78 ± 0.48 b	25.50 ***
Caffeic acid ^3^	12.63 ± 0.90 b	7.54 ± 0.39 a	7.90 0.32 a	5.12 *
Petunidin-3-*O*-glucoside ^1^	2.26 ± 0.12 a	4.75 ± 0.32 b	3.98 ± 0.41 ab	3.33 *
Peonidin-3-*O*-glucoside ^1^	1.07 ± 0.21 a	2.6 ± 0.30 b	2.44 ± 0.14 b	5.00 *
Polymeric pigments ^1^	43.73 ± 0.06 b	31.17 ± 0.09 a	38.99 ± 0.09 b	10.28 ***
Tannins ^2^	991.56 ± 46.28 b	639.65 ± 56.22 a	909.30 ± 44.39 b	11.29 ***
**Identity Profile Variables ^3^**	**Biodynamic**	**Conventional**	**Organic**	**F-Value**
Acetaldehyde	4.37 ± 0.31 a	9.81 ± 0.57 b	4.16 ± 0.37 a	5.16 *
2-Octanone	0.02 ± 0.01 a	0.02 ± 0.00 b	0.03 ± 0.00 b	6.08 *
Ethyl undecanoate	0.14 ± 0.14 b	0.07 ± 0.01 b	0.10 ± 0.02 a	11.52 ***
Ethyl acetate	115.28 ± 0.60 c	76.48 ± 0.73 a	103.91 ± 1.26 b	21.27 ***
Isoamyl acetate	0.86 ± 0.10 b	0.62 ± 0.05 a	0.78 ± 0.09 b	7.72 **
Diethyl succinate	3.12 ± 0.39 b	2.81 ± 0.16 ab	2.34 ± 0.18 b	4.03 *
Octanoic acid	0.33 ± 0.05 b	0.32 ± 0.06 b	0.19 ± 0.03 a	5.10 *

^1^ expressed as mg/L malvidin-3-*O*-glucoside; ^2^ expressed as mg/L (+)-catechin; ^3^ expressed as mg/L; * *p* ≤ 0.05; ** *p* ≤ 0.005; *** *p* ≤ 0.001.

**Table 5 foods-10-01894-t005:** Classification of the 2016 Chianti DOCG wines using SIMCA as a function of chemical and sensory variables (Eligibility, Identity, and All variables) for model development.

SIMCA Models		Chemical Classification			Sensory Classification		
		Eligibility	Identity	All Variables	Eligibility	Identity	All Variables
Conventional wines model	A_BD	•	•	•	•	−	•
	B_BD	•	•	•	•	−	•
	C_BD	•	−	•	−	−	•
	D_BD	•	•	−	•	−	•
	E_OR	−	•	•	−	•	•
	F_OR	•	•	•	−	−	−
	H_OR	•	•	•	−	•	•
	I_OR	•	•	•	•	−	−
Organic wines model	A_BD	•	•	•	•	•	•
	B_BD	•	•	•	•	•	•
	C_BD	•	−	•	−	•	•
	D_BD	•	•	•	•	•	•
	L_CV	•	•	•	•	•	•
	M_CV	•	•	•	•	•	•
	N_CV	•	•	•	•	−	−
	O_CV	•	•	•	•	•	•
	P_CV	•	•	•	•	•	•
Biodynamic wines model	E_OR	•	•	•	•	•	•
	F_OR	•	•	•	•	•	•
	H_OR	•	•	•	•	•	•
	I_OR	•	•	•	•	•	•
	L_CV	•	•	•	•	•	•
	M_CV	•	•	•	•	•	•
	N_CV	•	•	•	•	•	•
	O_CV	•	•	•	•	•	•
	P_CV	•	•	•	•	•	•

• Sample does fit the model; − Sample does not fit the model.

**Table 6 foods-10-01894-t006:** Typicality scores of Chianti DOCG wines assigned by the expert panel (mean value, standard deviation, statistical groups, and F-value). Different letters (a–f) indicate statistically significant differences (***: *p*-value < 0.001).

Wine	Typicality Scores
A_BD	3.20 ± 1.69 a
B_BD	3.71 ± 2.43 ab
C_BD	5.73 ± 2.63 f
D_BD	4.42 ± 2.28 bcd
E_OR	5.29 ± 2.56 ef
F_OR	4.16 ± 2.30 bc
H_OR	4.67 ± 2.46 cde
I_OR	5.04 ± 1.85 def
L_CV	4.89 ± 2.44 cdef
M_CV	4.98 ± 2.13 cdef
N_CV	5.44 ± 2.73 ef
O_CV	4.64 ± 2.34 cde
P_CV	5.58 ± 2.34 f
*F-Value*	5.69 ***

## Data Availability

Not applicable.

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
