# Peer review of "A Methodological Approach to Assess the Effect of Organic, Biodynamic, and Conventional Production Processes on the Intrinsic and Perceived Quality of a Typical Wine: The Case Study of Chianti DOCG"

_foods, 2021, doi:10.3390/foods10081894_

Round 1

Reviewer 1 Report

Interesting manuscript clearly showing the difficulties to find proper parameters to distinguish wines produced by different methodologies. 

There are some remarks on the manuscript.

1) Why are no data in the survey for producer A_BD not available?

2) Yield for producer L_CV with 55 kg/plant seems to be a typo, please correct it

3) Why are no metal analysis were conducted? Probably there is some additional information for separating the three different production methods

4) Please add the values for the standard edviation in tables 4 a&b and 6

Author Response

Here the answers to reviewer 1:

  • Why are no data in the survey for producer A_BD not available?

Thanks for the question. Unfortunately, A_BD data were not available and for this reason were not reported in the table.

  • Yield for producer L_CV with 55 kg/plant seems to be a typo, please correct it

Actually, it was a mistake and 55 was referred to q/ha of grape. We have corrected it with 1.57 kg/plant.

  • Why are no metal analysis were conducted? Probably there is some additional information for separating the three different production methods

Surely the elemental analysis could add important information about origin of wine and the management of the soil/plant in the vineyard. In our case, we were more oriented to study the chemical parameters that, according to literature, could be directly connected to the wine sensory characteristics.

Moreover, given those used for the study were all commercial wines, we have supposed that all elementals were below the maximum acceptable limits for which the OIV has set (i.e. Cu 1 mg/L max limit).

  • Please add the values for the standard edviation in tables 4 a&b and 6

As suggested, we have added the standard deviation.

Reviewer 2 Report

Overall speaking, I think this paper is well written. The introduction given the big picture of the topic and the focus; the differences among conventional, organic and biodynamic are well explained.  14 wine samples are organized and discussed in an understandable way; The chemical analysis data is well-collected and analyzed. However, the sensory analysis results, which might be the most important conclusion generated in the paper, did not convince readers since wine makers’ efforts are excluded in this article. Here are the reasons:

  1. Chianti Classico must contain at least 80% Sangiovese. In other words, the composition of 14 wines may share some similarity as well as dissimilarity in terms of wine grapes. These difference does not come from conventional, organic nor biodynamic; they came from the choice of wine makers. The composition of 14 wine does not included in the paper, which should be discussed.
  2. During the wine making process, barrel choices (new/old, origin of the barrel) and fermentation time makes the difference of the wine taste. These information does not include in the paper either. Instead, CO2 analysis is provided. As mentioned earlier, CO2 analysis is a good chemical analysis but not good indicator for sensory analysis. Wine prices can be another way to indicate the rough cost of whole wine making process.

There are many factors that will impact the taste of the wine, the article seems simplified the factors to wine growing and making using conventional, organic or biodynamic. More information about the wine making processes are needed to give a fair wine score comparison.       

Author Response

Here the answers to reviewer 2:

  1. Chianti Classico must contain at least 80% Sangiovese.In other words, the composition of 14 wines may share some similarity as well as dissimilarity in terms of wine grapes. These difference does not come from conventional, organic nor biodynamic; they came from the choice of wine makers. The composition of 14 wine does not included in the paper, which should be discussed.

The information about if the wine was produced as 100% Sangiovese or with a percentage of other red grape varieties blend (blend or not blend) were included in table 3 of the manuscript. As suggested, we have improved the description of the Chianti wines (Materials and Methods section and Results and discussions) and we have commented the data in table 3 related to the blend. The selected Chianti DOCG wines were all commercial and approved by the Consortium commission for their eligibility as Chianti. For this reason, the wines were supposed to be very similar between them because they belonged to the same kind of wine, but different for the winemaking style (that include the blend). Hence, we were interested to the systematic differences attributable to the kind of management and for this reason in the ANOVA we have used the factors management, judges and replicates in order to highlights the variability due to the kind of vineyard/winery management. In fact, in the main text, we highlighted the perceived differences attributable to the factor sample when discussing about the sensory profile (line 391-398).

  1. During the wine making process, barrel choices (new/old, origin of the barrel) and fermentation time makes the difference of the wine taste. These information does not include in the paper either. Instead, CO2 analysis is provided. As mentioned earlier, CO2 analysis is a good chemical analysis but not good indicator for sensory analysis. Wine prices can be another way to indicate the rough cost of whole wine making process.

There are many factors that will impact the taste of the wine, the article seems simplified the factors to wine growing and making using conventional, organic or biodynamic. More information about the wine making processes are needed to give a fair wine score comparison.     

The wines selected for the study belonged to the Chianti DOCG and usually this kind of product has not a long aging since it could be released at next March the 1st after the harvest. The choice to use the oak is admitted but if used, frequently it’s about very big wood barrel that in any case has a limited influence on the recognizability of wine, thus on wine typicality. In the present study, the winemaking practices applied by the different wineries were all reported in Table 3 and they were not systematically related to a particular kind of management.

The COanalysis was reported, not to relate it to the sensory profile, but to highlight the environmental impact of the different processes.

Concerning the wine price, it could be a very interesting aspect but we did not consider it since we were interested to chemical and sensory aspects.

Round 2

Reviewer 2 Report

My questions are answered.